# Thermoplastic Cellulose-Based Compound for Additive Manufacturing

**DOI:** 10.3390/molecules26061701

**Published:** 2021-03-18

**Authors:** Kirsi Immonen, Pia Willberg-Keyriläinen, Jarmo Ropponen, Asta Nurmela, Sini Metsä-Kortelainen, Otto-Ville Kaukoniemi, Heli Kangas

**Affiliations:** VTT Technical Research Centre of Finland Ltd., Tietotie 4E, P.O. Box 1000, FI-02044 VTT, FI-02150 Espoo, Finland; pia.willberg-keyrilainen@vtt.fi (P.W.-K.); jarmo.ropponen@vtt.fi (J.R.); asta.nurmela@vtt.fi (A.N.); sini.metsa-kortelainen@vtt.fi (S.M.-K.); otto-ville.kaukoniemi@vtt.fi (O.-V.K.); heli.kangas@vtt.fi (H.K.)

**Keywords:** thermoplastic cellulose, cellulose derivative, microcellulose, additive manufacturing, 3D printing, granule printing

## Abstract

The increasing environmental awareness is driving towards novel sustainable high-performance materials applicable for future manufacturing technologies like additive manufacturing (AM). Cellulose is abundantly available renewable and sustainable raw material. This work focused on studying the properties of thermoplastic cellulose-based composites and their properties using injection molding and 3D printing of granules. The aim was to maximize the cellulose content in composites. Different compounds were prepared using cellulose acetate propionate (CAP) and commercial cellulose acetate propionate with plasticizer (CP) as polymer matrices, microcellulose (mc) and novel cellulose-ester additives; cellulose octanoate (C8) and cellulose palmitate (C16). The performance of compounds was compared to a commercial poly(lactic acid)-based cellulose fiber containing composite. As a result, CP-based compounds had tensile and Charpy impact strength properties comparable to commercial reference, but lower modulus. CP-compounds showed glass transition temperature (Tg) over 58% and heat distortion temperature (HDT) 12% higher compared to reference. CAP with C16 had HDT 82.1 °C. All the compounds were 3D printable using granular printing, but CAP compounds had challenges with printed layer adhesion. This study shows the potential to tailor thermoplastic cellulose-based composite materials, although more research is needed before obtaining all-cellulose 3D printable composite material with high-performance.

## 1. Introduction

Due to the increased concern of environmental issues, novel renewable materials and additive manufacturing (AM) technologies are receiving more and more attention in several industries from medical to transportation industry [1,2]. In addition, the global and European level environmental policies are driving towards more sustainable and recyclable materials [3,4]. AM is considered a sustainable and material efficient technology, which uses only the material amount that is needed for the manufacturing of a part or a component, and the amount of waste is therefore low. AM has been used for decades for prototyping and during last years also widely for manufacturing of functional parts [5]. AM enables rapid and cost-effective manufacturing of complex and lightweight parts without any expensive tools such as molds.

AM technologies are divided under seven process categories according to basic principles of each process to create parts layer by layer [6]. The major differences between the process categories are the form of material e.g., powder, liquid, paste, sheets, wire or thermoplastic filaments or granules and the bonding agent or energy which are used in the construction of a part. Material extrusion systems, which are based on melting and dispensing of thermoplastic materials through a nozzle, are widely used in home 3D printing; however, there are also large-scale systems available mainly for industrial use [6]. Most of the home 3D printing systems use filaments, however there are many advantages to feed thermoplastic materials in a form of pellets or granules in industrial systems. These include high flow rates and big area prints, very large variety of materials and less expensive material processing costs as filament extrusion and spooling phases can be skipped [7,8,9].

The environmental issues and technology development have led towards increasing interest to develop novel sustainable high-performance materials for industrial additive manufacturing and for wide variation of applications such as electrical insulation components and parts for automotive and marine industries. Cellulose is a renewable and widely available material and is therefore an attracting alternative especially for fossil-based plastics; however, it is not thermoplastic by nature.

Cellulose is the most abundantly available biopolymer, and it covers up to 50 wt-% of lignocellulosic biomass [10,11]. Due to the cellulose inherent properties such as strong hydrogen bonding network, cellulose itself cannot be thermally processed and it is basically insoluble in traditional solvents. In order to impart thermoplastic properties to cellulose, its modification is needed, which is often challenging [12]. Currently well-known cellulose derivatives such as cellulose acetate (CA) and cellulose acetate propionate (CAP) are esterified through cellulose hydroxyl groups. Even though derivatization leads to increased thermoplastic properties their melt-processing window is narrow between melt flow temperature and degradation temperatures, especially with CA [13]. Therefore, quite large quantities of additives, typically plasticizers, are needed to obtain materials suitable for typical thermoplastic processes such as injection molding and extrusion. Long chain cellulose esters are bio-based cellulose esters with a side chain length of C6 or longer [14,15]. It has been reported earlier that for example cellulose octanoate (C8) and cellulose palmitate (C16) are melt processable without any additives [16,17]. These long alkyl side chains have a high plasticizing effect on the cellulose, suggesting that these thermoplastic cellulose compounds can also act as bio-based plasticizers in the composite materials.

The use of composite materials instead of neat polymers generally brings several advantages, such as improved stiffness and high specific strength [18]. Most of the thermoplastic composite materials for extrusion type additive manufacturing use fossil-based polymers such as ABS (acrylonitrile butadiene styrene), but increasing amounts of bio-based polymer PLA (polylactic acid) composites have entered to the market [19,20,21,22]. PLA is made from renewable raw material, originally from starch. It is an easy-to-print material however it is not very durable and does not stand high temperatures and UV light. The durability and other properties of ABS are clearly better than those of PLA, however ABS is not of renewable origin and releases non-pleasant fumes during printing. The stiffness properties of PLA-based composites are typically enhanced by adding different kind of fillers such as metal powder, carbon or glass-fibers, cellulose, microcellulose, wood and agro-fibers [23,24]. However, at the same time, tensile and impact strength properties are decreasing [23,25,26,27,28]. Also, due to quite low glass transition temperature, typically below 60 °C, the suitability of PLA-composite in many applications is limited. [29] Lignocellulose materials exhibit many interesting properties, including sustainability, hydrophilicity, biocompatibility, biodegradability, non-toxicity and broad chemical modifying capacity [24]. By using cellulose-based polymers, it is also possible to obtain materials with higher temperature resistance [30,31]. In medical applications extrusion printing of thermoplastic cellulose-derivatives for example ethyl cellulose, is a known technology [32,33]. To bring cellulose-based materials also in other applications is an interesting way to proceed.

Several different renewable fillers from continuous flax fibers to nanocellulose have been introduced to composite materials to bring their performance to a desired level. In thermoplastic materials, the typical improvements besides visual outlook and increased renewable material content are in improved material stiffness and specific strength [18]. The main fiber properties affecting composites are related to fiber type, size and aspect ratio. Typically, longer or continuous fibers and fibers with high aspect ratio provide better material properties. However, in material extrusion type manufacturing there are limitations to fiber size coming from printing nozzle, which narrows down the selection of suitable fillers. Microcellulose is one material, which due its size and shape is suitable for thermoplastic AM.

In this work, we studied the properties of thermoplastic cellulose-based composites and how they could be applied on novel 3D printing technique, printing of granules. Our aim was to maximize the cellulose content of the composite by using cellulose-based polymer matrix, novel cellulose-based additives as plasticizer and microcellulose to bring improved stiffness to the composite material. To evaluate the property level compared to existing commercial materials we used a commercial PLA-based cellulose containing composite material as a reference.

## 2. Results and Discussion

### 2.1. Test Series

Test series for additive manufacturing contained seven different compounds and two reference materials presented in Table 1. The following acronyms have been used in the coding: microcellulose (mc), commercial thermoplastic cellulose acetate prionate polymer with 17% commercial plasticizer (CP), thermoplastic cellulose esters; cellulose octanoate (C8) and cellulose palmitate (C16), cellulose acetate propionate without added plasticizer (CAP) and commercial printable PLA-based cellulose composite material reference (Cref).

All materials in Table 1 were injection molded to see the ultimate properties of the compounds without any effect originating from the 3D printing process. The 3D printing of materials was performed using granule printing (GP) process.

### 2.2. Results from the Printing Tests

Print head of the 3D printing system and granules used in the 3D printing are shown in the Figure 1. 3D printability of the material compositions was tested, while simultaneously manufacturing the specimens for tensile and impact strength tests. Descriptions of 3D printability of all the material compositions and images showing the 3D printing results and challenges are collected in Table 2. 3D printability described with “good” means that there was a constant material flow and good enough adhesion between the layers so that it was possible to manufacture 3D objects.

The results from the 3D printing trials with granule printing are showing good printability for reference commercial thermoplastic cellulose (CP), PLA-cellulose composite (Cref) and materials having commercial cellulose polymer with plasticizer (CP) as the polymer base. Most of the materials based on cellulose polymer without plasticizer (CAP) were 3D printable, however the 3D printed samples were very fragile. The fragile structure was a result of a poor adhesion between the 3D printed strings and layers. These materials with limited layer adhesion had an oily surface explored with fingers contrary to CP-materials with a drier and fibrous character. The most limited 3D printability was observed with CAP-mc-C8 and CAP-C16 of which there are extra images in the Table 2 showing the poor layer adhesion.

It is highlighted that in this study the main object was to compare different material compositions and to manufacture samples using 3D printing for further analysis and testing. For that reason, most of the 3D printing parameters were kept unchanged after a careful pre-testing of those. We believe that the properties of the 3D printed samples can be further enhanced by exploring and optimisation of the different 3D printing strategies and parameters in the future. Especially the quality of the 3D printing can be fine-tuned with cellulose composite materials having a good 3D printability.

### 2.3. Visual Results for Injection Molded Samples

All materials were injection molded to standard dog-bone shaped tests bars presented in Figure 2, that shows the visual difference between materials and some challenges caused by thermal stress.

The injection molded materials in Figure 2 are showing more clearly the inherent properties of the materials, for example color. It was relatively easy to produce the test specimens from all of the materials using injection molding. CP, CAP-C8 and CAP-C16 were all transparent. The dark color of the materials containing modified cellulose additive, C8, is partly coming from the brownish color of the additive itself. However, the two thermal stress cycles during compounding and injection molding at max 205 °C seem to cause some color formation, but probably not real degradation as their degradation temperature, presented in Table 3, is much higher than the processing temperatures. Also, the degradation of C8, explained in Section 2.5.1. occurs at higher temperature than used in injection molding. The cellulose-ester additive used in amounts of 4% in the mc containing CP compounds (CP-mc-C8, CP-mc-C16) lead to improved mc dispersion compared to CAP-mc-C8 and CAP-mc-C16 with 17% cellulose-ester additives. When comparing CP-mc and CP-mc-C16 the latter one had less color formation, probably due to higher plasticizer/additive content, 13% vs. 17%. This suggests that small addition of cellulose-ester, especially C16, has beneficial effect on mc dispersion and color formation in compound. Lightest color was in PLA-based Cref.

### 2.4. Scanning Electron Microscopy (SEM)

SEM-images of injection molded samples with 100× and 2500× enlargements are presented in Figure 3 and Figure 4, and 3D printed samples with 100× enlargement in Figure 5. The images are taken from the cross-cut surface in the middle of the test bar to see the dispersion of the fibers inside the material and to avoid the surface smoothing effect originating from the manufacturing process.

The SEM-images of the cross-cut of injection moulded materials with 100× enlargement in Figure 3 are showing the quite homogeneous structure of mc containing materials compared to Cref, where the the fibres are sticking out from the polymer structure.

SEM-images from the cross-cut of injection molded materials with 2500× enlargement in Figure 4 are showing how C8 and C16 addition, in amounts of 17%, on CAP makes the material flaky, if we compare it with CP, containing 13% of commercial additive. This might be due to some miscibility challenges of C8 and C16 with CAP. The longer the carbon chain in the cellulose additive is, the more flaky the compound with CAP is. This can be seen also in CAP-C8 and CAP-C16 images in Table 2, where it is described oiliness of these two materials and challenges of CAP-C16 layers to stick to each other. In mc containing materials with C8 and C16 (CP-mc-C8, CP-mc-C16, CAP-mc-C8 and CAP-mc-C16) the SEM-images of materials resemble each other and mc is quite tightly connected to cellulose polymer. Instead, in CP-mc, without the cellulose-based additive, mc is clearly visible as well as cellulose fibers in Cref, and both have visible gaps between fiber and polymer. The connection between the fiber and the polymer seems to be tighter with CP in CP-mc than with PLA in Cref.

SEM-images with 100× enlargement from the cross-cut of 3D-printed test bars in Figure 5 show the attachment between printed layers, but also significant amount of porosity inside the materials especially where fibers are included. The neat commercial polymer CP has good attachment between layers, but still some gaps in between the separate printed lines. CAP-C8 shows the same flaky cross-cut surface as with injection molded material. In the case of CAP-C16, the picture was taken from the side of the test bar due to total breakdown during sample preparation. Even this picture from the side of the CAP-C16 shows separation or poor attachment of the printed layers originating from the oily surface of the material containing 17% of cellulose-ester additive C16. However, C16 containing materials with mc (CP-mc-C16 and CAP-mc-C16) had both good printability as mentioned in Table 2. It is possible that mc absorbs part of the oily C16 and even improves mc dispersion inside the compound. In CP-mc-C16 the C16 amount was 4%, and thus significantly lower than 17% in the CAP-mc-C16, but probably still high enough to provide improved plasticization and fiber dispersion if compared to CP-mc without additional cellulose-ester additive. The layer adhesion, in general, was better in commercial polymer, CP-based compounds with 13% commercial plasticizer, as compared to CAP, where the only plasticizer was cellulose-ester additive in amounts of 17%. The layer adhesion was also good with commercial PLA-based compound Cref. The challenges in interlayer bond formation in 3D printing especially focused on filament printing, is discussed more widely for example by Lamm et al. [34]. However, similar regularities are relevant also in granular printing and maybe even in bigger scale, because there is one process step less, filament manufacturing.

### 2.5. Thermal Resistance Related Results

#### 2.5.1. Differential Scanning Calorimetry (DSC) and Thermogravimetric Analysis (TGA) Results

DSC analytics was made to see how the addition of mc, C8 and C16 affect the softening of the material (Tsoft), glass transition (Tg), melting (Tm) and crystallization (Tc) behavior of the composite materials. The material processing to products is going through two heating cycles, when granule printing is used (compounding and 3D-printing). By using DSC it is possible to see, if the material stabilizes within these heating cycles. TGA analytics was made to see, if the materials were stable in the processing temperatures and no degradation occurred. The DSC results for two heating runs and one cooling run are presented in Table 3 together with the TGA-results for the temperature where 5% of the material is degraded.

The results from the DSC-analysis indicate that some softening of the cellulose polymer-based materials starts already in temperatures above 34 °C during the first heating of the granule. This is connected to the mobility of the plasticizer in CP-based materials and softening of the plasticizing cellulose-ester additive side chains, C8 and C16, in CAP-based materials. The two melting peaks in CAP-based materials are related to additive melting and CAP-polymer melting (188–210 °C according to producer information). Typically, long chain thermoplastic cellulose-esters are amorphous materials, so their melting point is not clear. However, side chain crystallization and melting can occur when the length of the side chain is C10 or greater [35]. For cellulose palmitate (C16), side chain melting has been reported to be 20–35 °C depending on DS. [14,35,36] The two melting peaks in CAP based materials indicate poor miscibility of the thermoplastic cellulose-esters with CAP, that could be also seen in the SEM-image, Figure 4, for CAP-C8 and CAP-C16. The addition of cellulose-ester additive C8 and especially C16 is increasing amorphous character of compounds, which can be detected through smaller or disappearing crystalline changes during cooling of the material containing C8 or C16.

During the second heating the softening of CP-based materials in low temperatures seems to disappear and there are Tg’s close to 100 °C, that can indicate the stabilization of compounds during increasing heating cycles. Moreover, the one melting point indicates better misciblity of cellulose-based additives C8 and C16 with CP than with neat CAP. Compared to Cref, with Tg about 60 °C in both heating cycles, the cellulose-based compounds with mc seem to endure higher temperatures after second heating, which corresponds to readymade product after injection molding or extrusion manufacturing process.

TGA was analyzed to see if the materials can tolerate manufacturing temperatures up to 225 °C without significant degradation. According to results presented in Table 3 it is possible to use higher processing temperatures than normally suggested to cellulose fibers containing materials, below 200 °C. [18] Degradation temperatures of pure cellulose-based additives C8 and C16 have been reported to be between 230–270 °C [16] and are therefore clearly higher than the processing temperatures used in this research.

#### 2.5.2. Heat Distortion Temperature (HDT) Analysis

Heat distortion temperature (HDT) was analyzed for the injection molded materials to avoid the possible error coming from poor layer adhesion and to see the inherent properties coming only from the material. The results are presented in Table 4 and Figure 6.

Heat distortion temperature (HDT) is one of most important properties of polymeric materials indicating at what temperature the material starts to soften under a specific load. When comparing the HDT results in Table 4 and Figure 6 we can see that the commercial reference compound PLA-based Cref, has HDT value 55.4 °C and the commercial cellulose polymer CP (58 °C) and CP-mc (58.6 °C) are in the same level. For many applications those are quite low values. By adding cellulose-ester additive C16, the HDT was improved by 7% to 62.1 °C and it looks that C16 provides higher values compared to C8 cellulose-ester additive. When C16 is used as the only plasticizer/additive for CAP in CAP-C16, the HDT value was increased by 42% to 82.1 °C compared to CP. In mc containing compounds CAP-mc-C8 and CAP-mc-C16 the increase was 35% to 32% compared to Cref, respectively. By modifying CAP-based compound with different plasticizers, it seems possible to obtain materials with improved temperature resistance and that way to widen the applicability potential of thermoplastic cellulose composites.

### 2.6. Results from the Mechanical Tests

The mechanical strength tests such as tensile strength and Charpy impact strength values were analyzed for both injection molded materials and 3D granule printed test samples. The results for injection molded materials are showing the ultimate strength properties for each material. Results for 3D granule printed materials are reflecting also attachments of printed layers and internal porosity as well as fiber-polymer attachment of the material.

#### 2.6.1. Tensile Strength

Results from the tensile strength tests for injection molded and 3D printed materials are presented in Table 5. The separate results for tensile strength at yield are visualized in Figure 7, results for Young’s modulus in Figure 8 and results for strain at yield in Figure 9.

The tensile strength results in Table 5 and Figure 7 are in general reflecting the challenges between printed layer adhesion and pores by showing that during tensile stress the breakage of the sample occurs earlier than with injection molded samples. The strength results of printed samples are 10% to 78% lower than those of injection molded samples. The strength reduction in 3D printed and mc containing samples is the highest being 49% and 78% in CP-mc-C16 and CAP-mc-C16 respectively. Even in Cref the reduction is 56% from 26.6 MPa of injection molded to 11.6 MPa of 3D printed samples. The neat commercial polymer CP retained the strength best also in printed structure having the tensile strength value 23.6 MPa in printed samples and 26 MPa in injection molded samples, which is in-line with the relatively good adhesion between printed layers visible also in Figure 5. The low strength properties and porosity of 3D printed wood filled composite, but also their relation to printing parameters are reported also by Le Duigou et al. [37]. The best strength values for mc containing 3D printed material were obtained for CP-mc-C16, 12.9 MPa.

When focusing on injection molded samples, we can see that the tensile strength of all the 20–21% mc containing materials are at the level of 24 to 32 MPa. Even though CAP-C8 had some miscibility challenges, visible in Figure 4, by showing tensile strength 10.9 MPa, the 21% mc addition increased the strength value to 31.8 MPa, which may come from the partial impregnation of C8 to mc. However, the bigger miscibility challenges, visible in Figure 4, with C16 in CAP-C16 were not reflected to strength values of injection molded samples; that had tensile strength of 29 MPa but reduced to 24 MPa with mc addition.

The plasticization effect of C8 and C16 can be seen when comparing the tensile strength results of CP-mc, CP-mc-C8 and CP-mc-C16 having the strength values 28.9 MPa, 26.0 MPa and 25.5 MPa respectively.

The results for tensile modulus (Young’s) in Table 5 and Figure 8 are reflecting the same layer adhesion and porosity challenges as tensile strength results for 3D printed samples, as compared to injection molded, by showing up to 50% lower values for 3D printed fiber containing samples. The highest reduction is with CP-mc-C8 from 1670 MPa to 837 MPa. Modulus of the neat commercial polymer CP was in the same level with both manufacturing methods, if standard deviation is considered, 1360 MPa and 1297 MPa for injection molded and 3D printed respectively, provided by the good layer adhesion. 

In injection molded samples the mc addition to CP increases modulus in CP-mc by 26% from 1360 MPa to 1710 MPa. 4% C8 addition retained the modulus at the same level being 1670 MPa in CP-mc-C8, but 4% cellulose-ester C16 increased the modulus to 1840 MPa, which may be partially due to increased fiber dispersion. The modulus was also increased in CAP-based compounds after mc addition. In 17% C8 containing materials the mc addition caused 32% increase in modulus from 1539 MPa to 2034 MPa and in 17% C16 containing materials 18% increase from 1680 MPa to 1981 MPa. This modulus increase due to fiber material addition is in-line with previous studies [18].

The modulus of commercial PLA-based Cref with 20% fiber was 2642 MPa, thus being the stiffest in injection molded material, but still having the modulus reduction similar with the cellulose polymer-based material as 3D printed, when the modulus was 34% lower, 1754 MPa. Even though the layer adhesion, visible in Figure 5 was good, the 3D printed material contained quite a lot of porosity.

The elongation values for injection molded pure cellulose C16 and C8 are reported to be 12% and 50%, respectively [16]. These values are much higher than reported for commercially available shorter chain-length CAP [38]. By combining C8 and C16 with CAP and CP was assumed to achieve materials with higher elongation properties, which did not come true with these compounds. 

The strain at yield value for CP is quite same, 3.5–4%, for both injection molded and 3D printed samples due to good layer adhesion. The 21% mc and 4% cellulose-ester addition reduced the strain by dropping the strain value to level of 2.8–3.0%. C16 seems to have a slightly bigger reduction effect than C8. In 3D printed CP-based materials the reduction in strain was highest in CP-mc-C8 to 0.9% that may indicate fiber dispersion challenges. In CP-mc and CP-mc-C16, with strain 2.5%, fiber dispersion is probably good reflecting material internal elongation during stress. 

In CAP-based injection molded samples, the strain in 17% C16 containing material was 2.2% and clearly higher than in 17% C8 material, 0.8%, even though the elongation of C8 is reported to be higher. The difference may be due to some miscibility challenges between cellulose esters (C8 and C16) and CAP. C8 may be better compatible with CAP, when mc is added, which can be seen as strain at yield value for CAP-mc-C8, 2.1%. The 3D printed CAP-based materials showed all poor strain values due to poor layer adhesion as explained earlier. The strain at yield in 3D printed Cref was 22% lower than in injection molded material, 2.1% vs. 1.64%, indicating good fiber dispersion and quite good layer adhesion.

#### 2.6.2. Charpy Impact Strength

Charpy impact strength results for injection molded and 3D printed samples are presented in Table 6 and Figure 10. All the other results are for unnotched samples except CP, which did not break as unnotched and the test was made also as notched.

The Charpy impact strength (unnotched) results in Table 6 and Figure 10 are reflecting the brittle nature of the 3D printed mc containing commercial polymer CP containing samples, but also that of PLA-based cellulose fiber containing samples. As such the CP is a very tough material, which cannot be broken as unnotched, so the result is presented for notched samples. The high standard deviation in 3D printed sample is maybe due to the fact, that during notch preparation some detachment between printed layers may have occurred. 

The mc addition drops the impact strength result of the injection molded samples to the level of 21–22 kJ/m^2^ and 3D printed results to 5–6 kJ/m^2^ with or without 4% cellulose-ester (C8 or C16) addition. With CAP the 17% addition of cellulose-esters shows differences between C8 and C16 so that C16 is providing more impact ductility to the material than C8, which can be seen by comparing the results for injection molded samples CAP-C8, 4.5 kJ/m^2^, and CAP-C16, 15.2 kJ/m^2^. The impact strength of 3D printed samples from these two materials without fillers are surprisingly high when considering the loose layer adhesion visible in Figure 5, 8.4 kJ/m^2^ for CAP-C8 and 16 kJ/m^2^ for CAP-C16. However, during the test, the impact force needs to transfer from individual 3D printed string to another via only small contacts and not to transfer inside the even material, which may explain the impact results being even higher than for injection molded ones.

The mc addition to CAP containing 17% C16 dropped down the impact strength of injection molded material by 49% to 7.8 kJ/m^2^. With 17% C8 and mc the impact strength values were retained despite of the mc addition or even increased from 4.5 kJ/m^2^ to 6.8 kJ/m^2^. Also, in 3D printed samples with high cellulose-ester content (17%) and mc the impact strength was retained in the level of injection molded materials. That may also be due to similar force transfer effects between printed layers than in materials without mc.

The Charpy impact strength of injection molded PLA-based Cref was 18.8 kJ/m^2^, thus being better than with CAP-based materials, but 10–15% lower than in CP-based materials. As 3D printed the impact strength of Cref was dropped to 7.7 kJ/m^2^, indicating quite good layer adhesion, but high porosity and probably challenges in fiber polymer adhesion, which were also visible as gaps between fiber and polymer in Figure 4.

## 3. Materials and Methods

### 3.1. Materials

#### 3.1.1. Materials in Production of Thermoplastic Cellulose Additive

The cellulose for production of thermoplastic cellulose additives was commercial softwood dissolving grade pulp (Domsjö Fabriker AB, Sweden) with average Mw 520 kDa. The pulp was ozone pretreated according to a method described by Willberg-Keyriläinen et al. [35] to reduce the molar mass to 84 kDa. All other reagents were analytical grade and purchased from Sigma-Aldrich (Merck KGaA, Dramstadt, Germany).

#### 3.1.2. Materials in Manufacturing of Cellulose-Based Compound

Polymer matrices used for preparation of cellulose based compounds were cellulose acetate propionate without plasticizer (CAP) from Sigma-Aldrich (Merck KGaA, Darmstadt, Germany) with average Mn ~75,000, acetyl content 2.5 w-% and propionyl content 46 wt-%, and cellulose acetate propionate CELLIDOR CP300-13 (CP) (Albis Plastics GmbH, Hamburg, Germany) with phthalate free plasticizer content of 13% and melt flow rate 7.5 cm^3^/10 min (210 °C, 2.16 kg) [39,40]. Cellulose fiber used in compounds in amount of 20 wt-% was microcrystalline cellulose VIVAPUR 105 (JRS Pharma GmbH, Weissenborn, Germany) with average particle size by Laser diffraction 15 µm. In coupling of fiber and polymer was used reactive epoxidized linseed oil Lankroflex™ L (Valtris Specialty Chemicals, Independence, Ohio, USA). As additional plasticizer in compound in amounts of 4 wt-% and wt-17%, a thermoplastic cellulose-esters prepared at VTT according to method explained in chapter 3.2.1, were used. As commercial 3D printable reference material was used PLA-based compound containing 20% cellulose fiber (Cref).

### 3.2. Methods

#### 3.2.1. Preparation of Thermoplastic Cellulose Additive

Thermoplastic cellulose additives (cellulose octanoate (C8) and cellulose palmitate (C16) were prepared using the homogeneous method presented by Willberg-Keyriläinen et al. [35,41]. In this method, dry cellulose was first dissolved in a 5% LiCl/DMAc solution. Then fatty acid chloride (octanoyl chloride (C8) or palmitoyl chloride (C16), 3–4 equivalents to cellulose anhydroglucose unit (AGU) was added to the cellulose mixture using pyridine (3.6–4.6 equivalents/AGU) as catalyst. The reaction temperature was 80 °C and reaction time 16 hours. The cellulose esters were precipitated and washed with ethanol, C16 ester was additionally washed with acetone. The purity of these esters was determined using FTIR (Fourier-transform infrared spectroscopy) and ^1^H NMR (Nuclear Magnetic Resonance Spectroscopy) analyses, and no residues of free acid or fatty acid ethyl ester were detected after appropriate washing.

The degree of substitution (DS) of the prepared thermoplastic cellulose additives was analyzed using the solid state nuclear magnetic resonance (600 MHz NMR spectrometer, 10,000 scans, 10s recycle time, Agilent Technologies, Santa Clara, CA, USA) by comparing the cellulose esters carbonyl carbon (175 ppm) integrals with the cellulose C1 signal (105 ppm) integral. According to the NMR results, the DS values of thermoplastic cellulose C8 and cellulose C16 were 1.1 and 1.0, respectively.

#### 3.2.2. Processing of Thermoplastic Materials

Before compounding a coupling agent Lankroflex L was mixed with mc in amount of 5 % in relation to mc dry weight using a blade blender and the mixture was dried in 50 °C overnight. Also, thermoplastic cellulose additive C8 or C16 was dried in 50 °C overnight before compounding. The polymer matrices CAP and Cellidor CP300-13 (CP) were dried in 80 °C for 2 h.

Additive (Lankroflex L) containing mc and cellulose-ester were compounded with polymer matrices (CAP or CP-13) according to amounts presented in Table 1 using a co-rotating twin-screw extruder (Berstorff ZE 25x33 D, Berstorff GmbH, Hanover, Germany). The extruder zone temperatures ranged from 80 °C to 205 °C, speed 100 rpm and output 2 kg/h. The small pellets or granules from compounding were further processed using injection molding and 3D printing using granular printer explained in Section 3.2.3.

After compounding, the samples were injection molded with an injection-molding machine (Engel ES 200/50 HL, Engel Maschinenbau Geschellschaft m.b.H, Schwefberg, Austria) to test specimens according to ISO 527. The processing temperatures during injection molding were from 180 to 200 °C in the screw and 205 °C at nozzle and mold temperature 70 °C. The injection molding of Cref succeeded also in those temperatures.

#### 3.2.3. Additive Manufacturing

Samples for tensile and impact strength (see Section 3.2.6) testing were prepared using fused granular fabrication (FGF) AM system delivered by Brinter^®^. The system comprises a standard heated glass printing bed, tailored printing head for thermoplastic granules and system’s own software to control the movement of the printing head. BuildTak™ 3D printing sheets and glue stick were used on the top of the heated bed in order to increase the adhesion of the first layers.

Based on preliminary 3D printing trials, a printing speed of 15 mm/s and brass nozzles, type E3D-V6, with a diameter of 0.8 mm were selected and used in manufacturing of all samples. All samples were manufactured horizontally using a concentric infill pattern and 100% infill percentage. The number of layers were 4 and 8 for tensile and impact strength samples, respectively. The only exception was CAP-mc-C8 composition that had a very limited 3D printability and due to it the layer height of 0.25 mm was used and the number of layers was therefore double compared with the other material compositions. Temperatures of the nozzle and the bed and layer heights used in the manufacturing of the samples are collected in Table 7.

#### 3.2.4. Thermal Analytics

Thermo-gravimetric analytics (TGA) were performed using NETZSCH STA449F1 analyzer (NEZSCH GmbH, Selb, Germany). The TGA measurements were carried out in air atmosphere from 40 to 800 °C with a heating rate of 10 °C/min.

DSC (Differential scanning chromatography) analysis for compounds was made using NETZSCH DSC 204F1 Phoenix 240-12-0287-L (NEZSCH GmbH, Selb, Germany). The heating profile was two cycles at with 10°C/min heating and the cooling speed from −20 °C to 220 °C and back to −20 °C.

#### 3.2.5. Scanning Electron Microscopy (SEM)

The morphology of the injection molded, and 3D printed samples was studied using a SEM (scanning electron microscopy) from a cross-cut section of the test bar. The samples were cooled in liquid nitrogen and broken to two. The analysis was made from the broken surface. The sample surface was coated with gold to prevent surface charging. Gold film thickness on the sample surface was 50–70 nm. Analyses were made using JEOL JSM T100 (JEOL ltd., Tokyo, Japan) with a voltage of 25 kV.

#### 3.2.6. Mechanical Tests

Tensile tests were performed according to ISO-527 standard using an Instron 4505 Universal Tensile Tester (Instron Corp., Canton, MA, USA) and an Instron 2665 Series High Resolution Digital Automatic Extensometer (Instron Corp., Canton, MA, USA) with a 10 kN load cell and a 5 mm/min cross-head speed. Five parallel specimens were tested for each sample material to obtain the average values of the tensile properties.

Impact strengths were determined according to ISO-179 standard using a Charpy Ceast Resil 5.5 Impact Strength Machine (CEAST S.p.a., Torino, Italy). Charpy impact strength tests in edgewise orientation was applied to the unnotched specimens. Impact testing was made for six to ten replicates to obtain an average value and variance.

The test specimens were kept in standard conditions (23 °C, 50% relative humidity) for at least five days before testing.

#### 3.2.7. Heat Distortion Temperature (HDT)

Heat distortion temperature (HDT) was measured according to the ISO-75 standard using method A with 1.8. MPa stress on the sample. HDT was determined by using the Ceast HDT 3 VICAT P/N 6911.000 (Ceast S.p.a., Torino, Italy) and for three parallel samples.

## 4. Conclusions

The aim of this work was to study the properties of thermoplastic cellulose-based composites and how they could be applied on a novel 3D printing technique, printing of granules. The focus was to maximize the cellulose content of the composite by using two cellulose-based polymer matrices; neat cellulose acetate propionate (CAP) polymer and commercial cellulose acetate propionate containing 13% commercial plasticizer (CP), two novel cellulose-ester additives and microcellulose (mc) to bring improved stiffness to the composite material. The novel cellulose-ester additives were cellulose octanoate (C8) and cellulose-palmitate (C16), and their function as fiber dispersing and CAP plasticizer additive was analyzed. To evaluate the property level of the developed cellulose-based composites compared to existing commercial materials a commercial PLA-based cellulose fiber containing composite material was used as a reference (Cref). Materials were also injection molded to see their best possible material properties.

It was possible to manufacture all the materials for test samples using 3D granular printer, however this manufacturing method revealed clearly the differences between the material combinations. The commercial CP, containing 13% commercial plasticizer, was as such printable and showed good strength properties both as 3D printed and injection molded. Also, CP-based, mc containing composite was printable with and without 4% cellulose-ester addition and showed adequate adhesion between the printed layers, but also high porosity inside the material. The more experimental, CAP-based materials with 17% cellulose-esters had challenges in printed layer adhesion and the surface of compounds were oily indicating that the amount of cellulose esters, 17%, was too high to mix properly with CAP. Also, the mc-containing CAP compounds showed high porosity inside the material in 3D printed samples. Porosity was discovered also in the 3D printed reference PLA-based Cref. The high porosity reflects the challenges within granular printing method. However, the main objective of this work was to compare different material compositions and to manufacture samples using 3D printing for further analysis and testing. For that reason, most of the 3D printing parameters were kept unchanged after a careful pre-testing of those. We believe that the properties of the 3D printed samples can be further enhanced by optimization of the 3D printing parameters for the selected material in the future.

The tensile strength properties of the injection molded CP-materials and Cref were comparable to each other having tensile strength between 25.5 to 29 MPa. The Charpy impact strength in CP-compounds was 12–20% higher compared to Cref, 18.8 kJ/m^2^. However, the tensile modulus of Cref is 30–37% higher compared to CP-compounds, CP-mc-C8 had modulus of 1670 MPa, CP-mc 1720 MPa and CP-mc-C16 1840 MPa. Regarding the CAP-mc compounds, there is potential for higher tensile strength and modulus with C8 addition. CAP-mc-C8 had tensile strength of 31.8 MPa and modulus of 2034 MPa, but quite low impact strength, 6.8 kJ/m^2^, thus showing the brittleness of the material. The cellulose-ester, C8 and C16, materials showed some miscibility challenges with CAP, especially in high contents, 17%, which seemed to be too much. However, there was indication of improved fiber (mc) dispersion when lower cellulose-ester amounts, 4%, were used.

The thermal behavior of the PLA-based Cref showed very constant Tg of 60 °C in two heating cycles and HDT temperature 55.4 °C. The commercial CP-based materials stabilized after the first heating cycle in DSC-analysis showing the Tg of 95–103 °C during the second heating, depending on the compound. CP-mc-C16 had the highest HDT-value, 62.1 °C, of the CP-based materials but with CAP and C16 (17%) as the only additive, HDT-value of 82.1 °C was achieved.

This study indicates a potential to tailor high temperature resistant cellulose composite materials using thermoplastic cellulose acetate propionate as the polymer matrix material in combination with microcellulose and possibly with small amounts of novel cellulose-esters as fiber dispersing additive. More studies are still needed to optimize the additive and filler amounts, as well as cellulosic polymer base towards all-cellulose composites competitive with PLA-based commercial composites. Also, optimization for printing parameters is needed, if the focus is 3D printable materials and future manufacturing technologies.

## Figures and Tables

**Figure 1 molecules-26-01701-f001:**
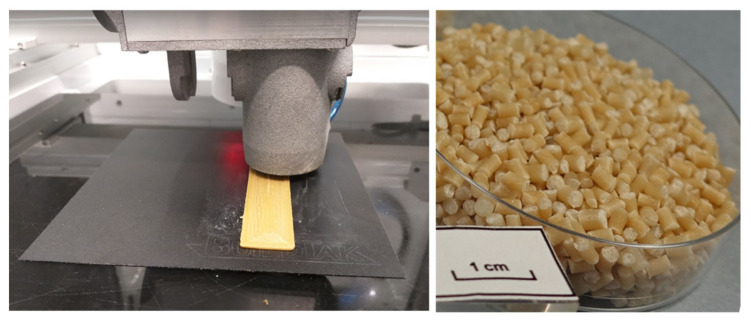
Granulate 3D printing head (**left**) and granules used in 3D printing process (**right**).

**Figure 2 molecules-26-01701-f002:**
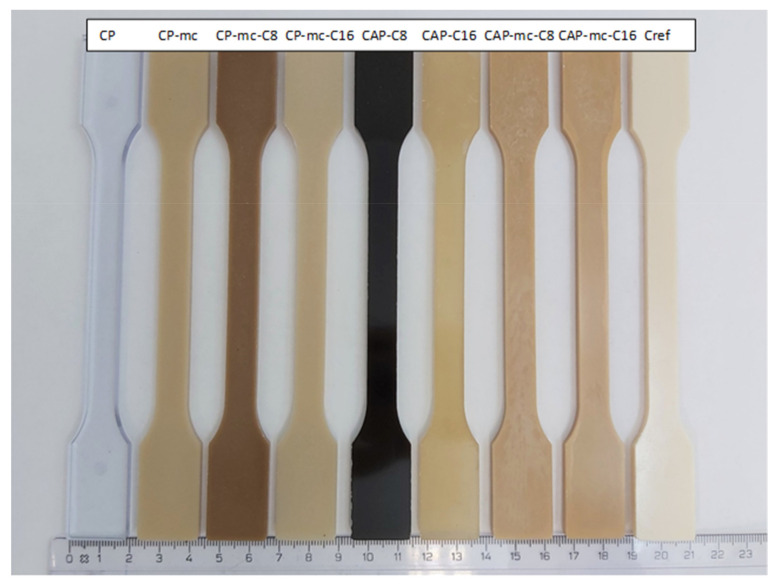
Injection molded test bars.

**Figure 3 molecules-26-01701-f003:**
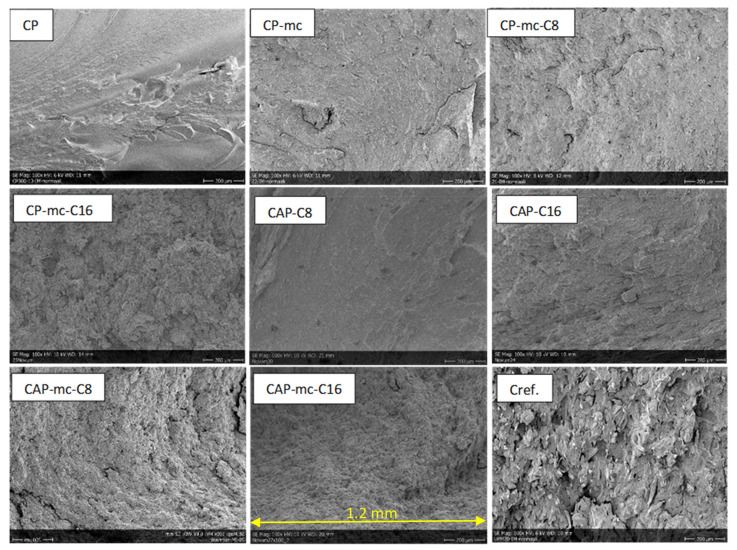
SEM-pictures with 100× enlargement from cross-cut middle of the injection molded test bars. The width of each pictures is 1.2mm. CP is commercial cellulose acetate propionate with 13% commercial plasticizer; CP-mc is 20% microcellulose containing CP; CP-mc-C8 contains also 4% cellulose octanoate C8; CP-mc-C16 contains 4% of cellulose palmitate C16; CAP-C8 contains neat cellulose acetate propionate (CAP) with 17% C8; CAP-C16 contain 17% C16 in CAP; CAP-mc-C8 contains 20% mc and 17% C8 in CAP; CAP-mc-C16 contains 20% mc and 17% C16 in CAP and Cref is PLA-based material with 20% cellulose fibre.

**Figure 4 molecules-26-01701-f004:**
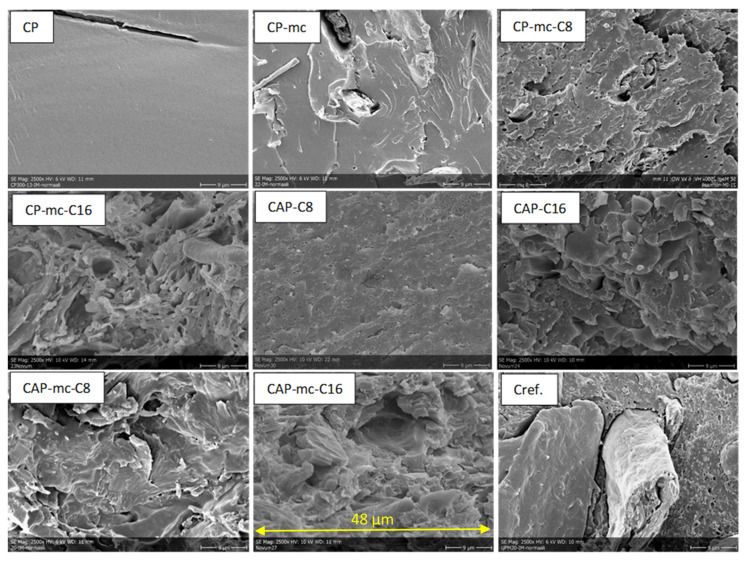
SEM-pictures with 2500× enlargement from cross-cut middle of the injection molded test bars. The width of each pictures is 48 µm. CP is commercial cellulose acetate propionate with 13% commercial plasticizer; CP-mc is 20% microcellulose containing CP; CP-mc-C8 contains also 4% cellulose octanoate C8; CP-mc-C16 contains 4% of cellulose palmitate C16; CAP-C8 contains neat cellulose acetate propionate (CAP) with 17% C8; CAP-C16 contain 17% C16 in CAP; CAP-mc-C8 contains 20% mc and 17% C8 in CAP; CAP-mc-C16 contains 20% mc and 17% C16 in CAP and Cref is PLA-based material with 20% cellulose fibre.

**Figure 5 molecules-26-01701-f005:**
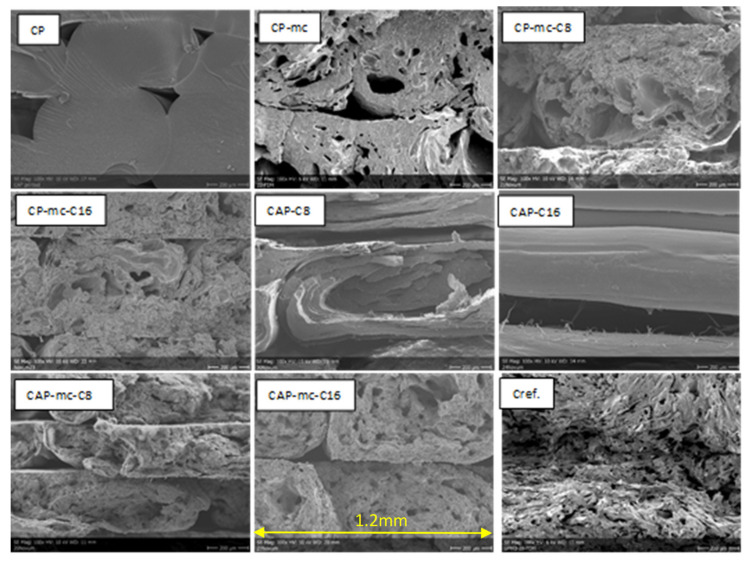
SEM-pictures with 100× enlargement from the cross-cut of 3D printed test bars. CAP-C16 is from the side of the printed test bar. The width of each pictures is 1.2 mm. CP is commercial cellulose acetate propionate with 13% commercial plasticizer; CP-mc is 20% microcellulose containing CP; CP-mc-C8 contains also 4% cellulose octanoate C8; CP-mc-C16 contains 4% of cellulose palmitate C16; CAP-C8 contains neat cellulose acetate propionate (CAP) with 17% C8; CAP-C16 contain 17% C16 in CAP; CAP-mc-C8 contains 20% mc and 17% C8 in CAP; CAP-mc-C16 contains 20% mc and 17% C16 in CAP and Cref is PLA-based material with 20% cellulose fibre.

**Figure 6 molecules-26-01701-f006:**
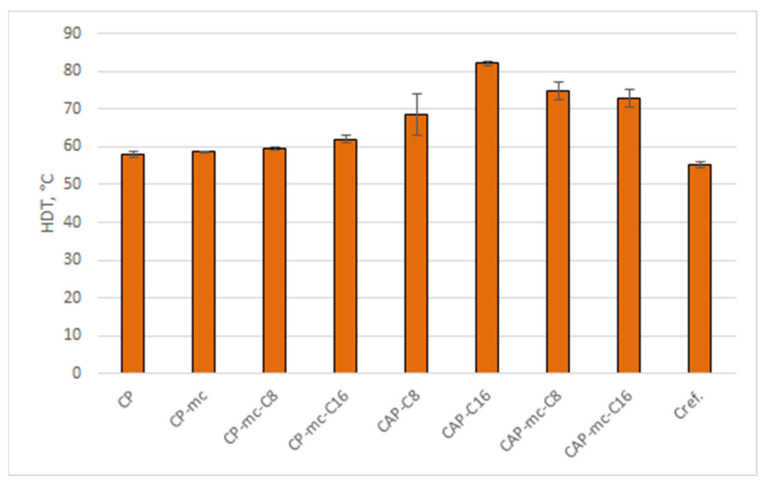
Heat distortion temperatures for injection molded materials. CP is commercial cellulose acetate propionate with 13% commercial plasticizer, CP-mc is 20% microcellulose containing CP, CP-mc-C8 contains also 4% cellulose octanoate C8, CP-mc-C16 contains 4% of cellulose palmitate C16, CAP-C8 contains neat cellulose acetate propionate (CAP) with 17% C8, CAP-C16 contain 17% C16 in CAP, CAP-mc-C8 contains 20% mc and 17% C8 in CAP, CAP-mc-C16 contains 20% mc and 17% C16 in CAP and Cref is PLA-based material with 20% cellulose fibre.

**Figure 7 molecules-26-01701-f007:**
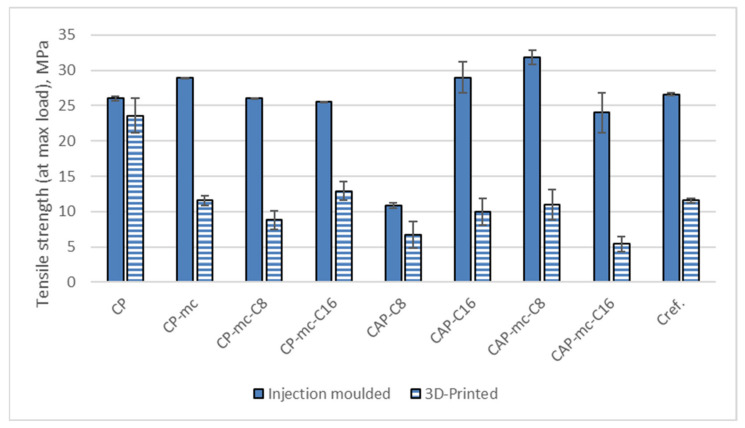
Tensile strength at yield for injection molded and 3D printed materials. CP is commercial cellulose acetate propionate with 13% commercial plasticizer; CP-mc is 20% microcellulose containing CP; CP-mc-C8 contains also 4% cellulose octanoate C8; CP-mc-C16 contains 4% of cellulose palmitate C16; CAP-C8 contains neat cellulose acetate propionate (CAP) with 17% C8; CAP-C16 contain 17% C16 in CAP; CAP-mc-C8 contains 20% mc and 17% C8 in CAP; CAP-mc-C16 contains 20% mc and 17% C16 in CAP and Cref is PLA-based material with 20% cellulose fibre.

**Figure 8 molecules-26-01701-f008:**
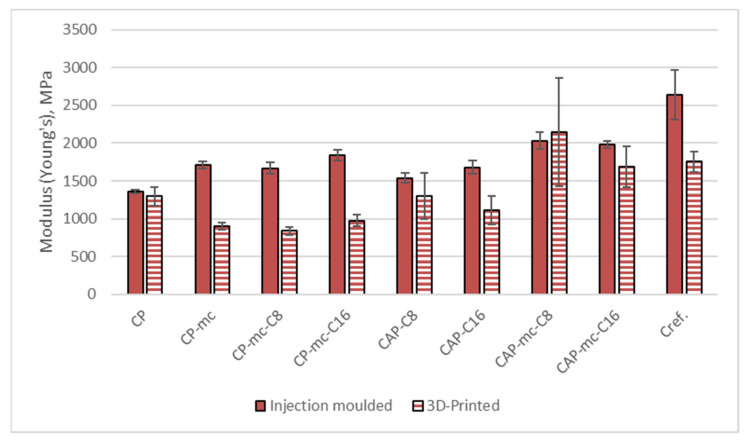
Young’s modulus for injection molded and 3D printed materials. CP is commercial cellulose acetate propionate with 13% commercial plasticizer; CP-mc is 20% microcellulose containing CP; CP-mc-C8 contains also 4% cellulose octanoate C8; CP-mc-C16 contains 4% of cellulose palmitate C16; CAP-C8 contains neat cellulose acetate propionate (CAP) with 17% C8; CAP-C16 contain 17% C16 in CAP; CAP-mc-C8 contains 20% mc and 17% C8 in CAP; CAP-mc-C16 contains 20% mc and 17% C16 in CAP and Cref is PLA-based material with 20% cellulose fibre.

**Figure 9 molecules-26-01701-f009:**
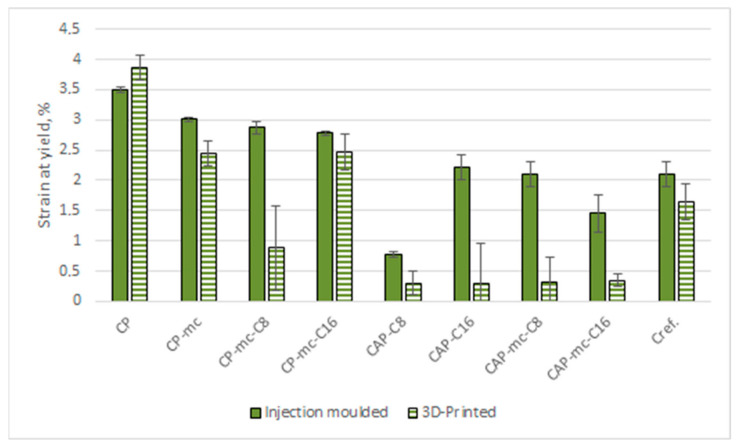
Strain at yield for injection molded and 3D printed materials. CP is commercial cellulose acetate propionate with 13% commercial plasticizer; CP-mc is 20% microcellulose containing CP; CP-mc-C8 contains also 4% cellulose octanoate C8; CP-mc-C16 contains 4% of cellulose palmitate C16; CAP-C8 contains neat cellulose acetate propionate (CAP) with 17% C8; CAP-C16 contain 17% C16 in CAP; CAP-mc-C8 contains 20% mc and 17% C8 in CAP; CAP-mc-C16 contains 20% mc and 17% C16 in CAP and Cref is PLA-based material with 20% cellulose fibre.

**Figure 10 molecules-26-01701-f010:**
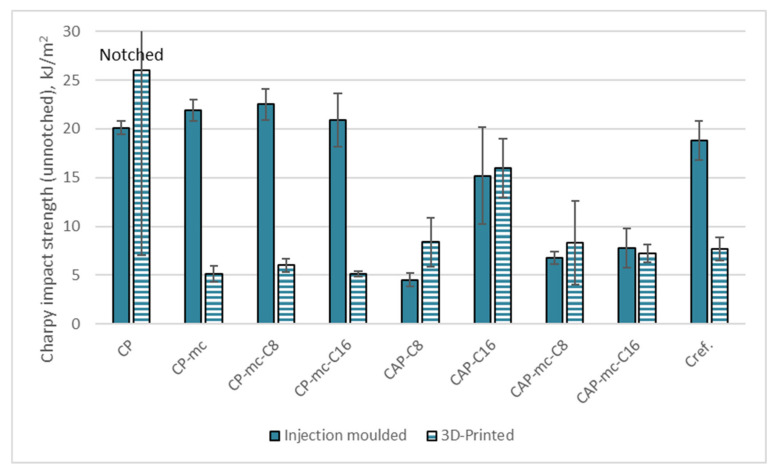
Charpy impact strength results for injection molded and 3D printed materials. Unnotched samples except CP, which did not break as unnotched. CP is commercial cellulose acetate propionate with 13% commercial plasticizer; CP-mc is 20% microcellulose containing CP; CP-mc-C8 contains also 4% cellulose octanoate C8; CP-mc-C16 contains 4% of cellulose palmitate C16; CAP-C8 contains neat cellulose acetate propionate (CAP) with 17% C8; CAP-C16 contain 17% C16 in CAP; CAP-mc-C8 contains 20% mc and 17% C8 in CAP; CAP-mc-C16 contains 20% mc and 17% C16 in CAP and Cref is PLA-based material with 20% cellulose fibre.

**Table 1 molecules-26-01701-t001:** Test series of compounds.

Code	Thermoplastic Cellulose Ester (C) Type	Thermoplastic Cellulose Ester (C) Content, (%)	Cellulose Fibre Content, (%)	Description
CP		0	0	Commercial thermoplastic cellulose polymer
CP-mc		0	20	
CP-mc-C8	C8	4	20	
CP-mc-C16	C16	4	20	
CAP-C8	C8	17	0	CAP without commercial plasticizer
CAP-C16	C16	17	0	CAP without commercial plastizicer
CAP-mc-C8	C8	17	20	CAP without commercial plastizicer
CAP-mc-C16	C16	17	20	CAP without commercial plastizicer
Cref		0	20	Commercial cellulose fibre containing reference material

**Table 2 molecules-26-01701-t002:** Results from 3D printing trials with granule printing method.

Code	3D printability	Image
CP	Good	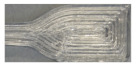
CP-mc	Good	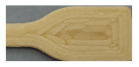
CP-mc-C8	Good	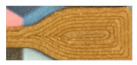
CP-mc-C16	Good	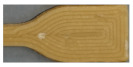
CAP-C8	Good, however poor layer adhesion, oily material, fragile	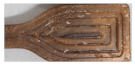
CAP-C16	Limited, however poor layer adhesion, oily material, fragile	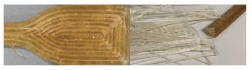
CAP-mc-C8	Limited, and poor layer adhesion, oily material	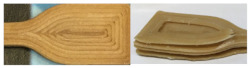
CAP-mc-C16	Good, however poor layer adhesion, oily material, fragile	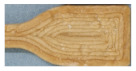
Cref	Good	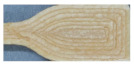

**Table 3 molecules-26-01701-t003:** Results from DSC and TGA analytics. DSC-results are from the first heating, cooling and second heating cycles for the compounded granules. TGA results are for the temperature, where 5% of the material is degraded.

Code	First Heating		Cooling	Second Heating		TGA, 5% Degraded
	Tsoft, °C	Tm, °C	Tc, °C	Tg, °C	Tm, °C	Tdeg, °C
CP	42	158	122	95	159	267
CP-mc	47	157	124	97	160	273
CP-mc-C8	40	161	123	103	160	264
CP-mc-C16	34	114/161	n.d.	103	163	277
CAP-C8	54	173/191	n.d.	55	154/181	267
CAP-C16	36	173/206	n.d.	32	146/172	298
CAP-mc-C8	72	161/182	145	78	142/194	309
CAP-mc-C16	34	98/168	n.d.	84	175/194	305
Cref	61	149	125	60	165	304

Standard deviation in DSC-method: ± 2 °C (DIN 53765).

**Table 4 molecules-26-01701-t004:** HDT-results for injection molded materials.

Code	HDT, °C
CP	58 ± 0.8
CP-mc	58.6 ± 0.3
CP-mc-C8	59.6 ± 0.3
CP-mc-C16	62.1 ± 0.9
CAP-C8	68.6 ± 5.6
CAP-C16	82.1 ± 0.7
CAP-mc-C8	74.8 ± 2.3
CAP-mc-C16	72.9 ± 2.4
Cref	55.4 ± 0.8

**Table 5 molecules-26-01701-t005:** Results from tensile strength test for injection molded and 3D printed materials.

	Injection Moulded			Granule Printed		
	Strength at Yield, MPa	Young’s Modulus, MPa	Strain at Yield, %	Strength at Yield, MPa	Young’s Modulus, MPa	Strain at Yield, %
CP	26.0 ± 0.3	1360 ± 25	3.5 ± 0.04	23.6 ± 2.5	1297 ± 121	3.9 ± 0.2
CP-mc	28.9 ± 0.1	1710 ± 45	3.0 ± 0.04	11.6 ± 0.7	902 ± 42	2.5 ± 0.2
CP-mc-C8	26.0 ± 0.1	1670 ± 74	2.9 ± 0.1	8.8 ± 1.3	837 ± 54	0.9 ± 0.7
CP-mc-C16	25.5 ± 0.1	1840 ± 69	2.8 ± 0.04	12.9 ± 1.3	977 ± 81	2.5 ± 0.3
CAP-C8	10.9 ± 0.4	1539 ± 65	0.8 ± 0.04	6.7 ± 1.0	1297 ± 307	0.3 ± 0.2
CAP-C16	29.0 ± 2.2	1680 ± 87	2.2 ± 0.2	10.0 ± 1.9	1115 ± 185	0.3 ± 0.7
CAP-mc-C8	31.8 ± 1.0	2034 ± 116	2.1 ± 0.2	11.0 ± 2.1	2147 ± 717	0.3 ± 0.4
CAP-mc-C16	24.0 ± 2.8	1981 ± 51	1.5 ± 0.3	5.4 ± 1.1	1690 ± 274	0.4 ± 0.1
Cref	26.6 ± 0.2	2642 ± 329	2.1 ± 0.2	11.6 ± 0.3	1754 ± 133	1.6 ± 0.3

**Table 6 molecules-26-01701-t006:** Charpy impact strength results for injection molded and 3D printed materials. Unnotched samples except CP, which did not break as unnotched.

Code	Injection Moulded	3D Printed
	kJ/m^2^	kJ/m^2^
CP	no break unnotched 20.1 ± 0.7 notched	no break unnotched 26.0 ± 19.0 notched
CP-mc	21.9 ± 1.1	5.1 ± 0.8
CP-mc-C8	22.5 ± 1.6	6.0 ± 0.7
CP-mc-C16	20.9 ± 2.7	5.1 ± 0.3
CAP-C8	4.5 ± 0.7	8.4 ± 2.5
CAP-C16	15.2 ± 5.0	16 ± 3.0
CAP-mc-C8	6.8 ± 0.6	8.3 ± 4.3
CAP-mc-C16	7.8 ± 2.0	7.2 ± 0.9
Cref	18.8 ± 2.0	7.7 ± 1.2

**Table 7 molecules-26-01701-t007:** Parameters used in 3D printing trials.

Code	Temperature Nozzle/Bed (°C)	Layer Height (mm)
CP	200/60	0.5
CP-mc	215/60	0.5
CP-mc-C8	210/60	0.5
CP-mc-C16	219/60	0.5
CAP-C8	185/60	0.5
CAP-C16	220/60	0.5
CAP-mc-C8	220/60	0.25
CAP-mc-C16	224/60	0.5
Cref	215/60	0.5

## Data Availability

Not applicable.

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
