# Peer review of "Thermoplastic Cellulose-Based Compound for Additive Manufacturing"

_molecules, 2021, doi:10.3390/molecules26061701_

Round 1
Reviewer 1 Report
This work is particularly interesting for researchers working in the field of bio-based plastic materials.
Although this manuscript is well written and relevant in most of its interpretations, it lacks, in my opinion, some essential information to understand and interpret the results correctly and to confirm the conclusions made by the authors.
- The authors used 2 different cellulose acetate propionate matrices, one with a plasticizer (phtalate), the other without plasticizer. But do the two matrices have the same characteristics in terms of molar mass but also acetyl and propionyl ratio? This can have a real influence on the characteristics, in particular mechanical, of the reference products but also on FACEs-added products (FACE = fatty acid cellulose ester).
- In the coupling of the fiber and the polymer, the authors use a commercial reactive epoxidized linseed oil (Lankroflex™). However, this coupling agent can have, because of its origin (vegetal oil), a plasticizing effect on the polymer matrix itself. It would have been advisable to carry out tests on the CPs and CAPs by simply adding 5% of this coupling agent in order to study its influence on the properties of the polymer. If LankroflexTM influences the properties of microcrystalline cellulose unloaded materials, this can distort results, especially mechanical ones.
- The C8 and C16 FACEs were synthesized by the method of Willberg-Keyriläinen et al, which is one of the methods reported in the literature for homogeneous modification of cellulose with fatty acid chlorides. This synthesis, in the present manuscript, should be more described, in particular the part "purity" of the FACEs. In fact, the process used leads, during the purification, to the formation of fatty acid ethyl esters which, if not removed, can also play a role in the plasticization mechanism of the tested materials. A simple 1H or 13C NMR analysis can be used to see the presence (or not) of the characteristic signals of the protons (or carbons) of the corresponding ethyl group.
- The most important problem is the lack of information regarding the DS of the FACES used. Indeed, depending on these DS, the mechanical and thermal properties of the FACEs are different, and will therefore have an impact on the properties of the final material. With the length of the fatty chain, the DS is the 2nd crucial parameter which must be particularly well-defined when working with FACEs, and this is not the case here. And in the case of this job, different DS would be a serious problem
- The second major problem is the lack of information concerning the thermal characteristics of FACEs. Indeed, if those of commercial CP and CAP products are well established and re-evaluated in this work, it would have been relevant to evaluate the characteristics of FACEs by DSC and TGA in order to know their glass transition temperatures, melting and degradation temperatures. The determination of these temperatures is also essential to study the thermal processability of materials based on FACEs... Recent data can be found in the literature for DSC analyzes according to DS and fatty acid length (Carbohydrate Polymers 234 (2020) 115912) or for the thermogravimetry data of FACEs (ex: J. Appl. Poly. Sci., 2006, 100 (2), 1093-1102). This missing information should be provided before considering publication of this manuscript.
- The explanation on the "burnt" appearance of the CAP-C8 tensile test specimens should be reviewed (page 5, figure 2 and text below), in accordance with Point 5 raised just before, and taking into account the working temperatures for other composites using C8 FACE. If the thermogravimetric analyzes had been carried out on the FACEs before injection, this could have been anticipated by reducing (slightly) the injection temperature for example, below the degradation temperature of the C8 FACE degradation (if it is indeed a degradation of the C8 FACE of course)
According to these remarks, I recommend major revisions before considering this manuscript for publication in Molecules
Author Response
Reviewer 1
This work is particularly interesting for researchers working in the field of bio-based plastic materials.
Although this manuscript is well written and relevant in most of its interpretations, it lacks, in my opinion, some essential information to understand and interpret the results correctly and to confirm the conclusions made by the authors.
- The authors used 2 different cellulose acetate propionate matrices, one with a plasticizer (phtalate), the other without plasticizer. But do the two matrices have the same characteristics in terms of molar mass but also acetyl and propionyl ratio? This can have a real influence on the characteristics, in particular mechanical, of the reference products but also on FACEs-added products (FACE = fatty acid cellulose ester).
Answer: This is a good comment. Unfortunately, the manufacturer, Albis Plastics, gives no information about molar mass or acetyl or propionyl ratio for the commercial engineering type cellulose acetate propionate (CAP) Cellidor CP300-13 containing 13% phtalate-free plasticizer. CAP from Sigma-Aldrich has molar mass Mn ~75000 and contains ~2.5% acetyl and ~46% propionyl. Because all the properties of matrices were not known, we harmonized the amounts of plasticizers or plasticizing additives in the compounds in relation to commercial additive amount.
2. In the coupling of the fiber and the polymer, the authors use a commercial reactive epoxidized linseed oil (Lankroflex™). However, this coupling agent can have, because of its origin (vegetal oil), a plasticizing effect on the polymer matrix itself. It would have been advisable to carry out tests on the CPs and CAPs by simply adding 5% of this coupling agent in order to study its influence on the properties of the polymer. If LankroflexTM influences the properties of microcrystalline cellulose unloaded materials, this can distort results, especially mechanical ones.
Answer: Thank you, this is very relevant comment. The authors were aware of the plasticizing effect of Lankroflex™ L. It was used 5% in relation to fibre amount, which means 1% in total compound. It was also mixed and impregnated in advance with microcellulose and dried with microcellulose overnight in 50°C. So, the authors assumed that the plasticizing effect on polymer matrix would be minimal compared to the plasticizing effect of FACEs and commercial plasticizer used in much higher amounts up to 17%. The main focus with this additive use was to minimize the error coming from the poor fiber-polymer connection, but yes, then there is the question of this plasticizing effect.
3. The C8 and C16 FACEs were synthesized by the method of Willberg-Keyriläinen et al, which is one of the methods reported in the literature for homogeneous modification of cellulose with fatty acid chlorides. This synthesis, in the present manuscript, should be more described, in particular the part "purity" of the FACEs. In fact, the process used leads, during the purification, to the formation of fatty acid ethyl esters which, if not removed, can also play a role in the plasticization mechanism of the tested materials. A simple 1H or 13C NMR analysis can be used to see the presence (or not) of the characteristic signals of the protons (or carbons) of the corresponding ethyl group.
Answer: Thank you for this comment. The purity of these esters was determined using FTIR and 1H NMR analyses, and no residues of free acid or fatty acid ethyl ester were detected after appropriate washing. This purity determination has now been added to the section 3.2.1.
4. The most important problem is the lack of information regarding the DS of the FACES used. Indeed, depending on these DS, the mechanical and thermal properties of the FACEs are different, and will therefore have an impact on the properties of the final material. With the length of the fatty chain, the DS is the 2nd crucial parameter which must be particularly well-defined when working with FACEs, and this is not the case here. And in the case of this job, different DS would be a serious problem
Answer: Thank you for this comment. We regret that the DS data of the FACEs has been forgotten to mention. DS is a very important parameter that significantly affects the properties of the material. According to the NMR results, the DS values of thermoplastic cellulose C8 and cellulose C16 were 1.1 and 1.0, respectively. We have now added this information to the section 3.2.1.
5. The second major problem is the lack of information concerning the thermal characteristics of FACEs. Indeed, if those of commercial CP and CAP products are well established and re-evaluated in this work, it would have been relevant to evaluate the characteristics of FACEs by DSC and TGA in order to know their glass transition temperatures, melting and degradation temperatures. The determination of these temperatures is also essential to study the thermal processability of materials based on FACEs... Recent data can be found in the literature for DSC analyzes according to DS and fatty acid length (Carbohydrate Polymers 234 (2020) 115912) or for the thermogravimetry data of FACEs (ex: J. Appl. Poly. Sci., 2006, 100 (2), 1093-1102). This missing information should be provided before considering publication of this manuscript.
Answer: We have now added information on the thermal properties of FACEs to section 2.5.1. Typically, FACEs are amorphous materials, so their melting point is not clear. However, side chain crystallization and melting can occur when the length of the side chain is C10 or greater. For cellulose C16, side chain melting has been earlier reported to be 20-35 °C depending on DS. Degradation temperatures of pure cellulose C8 and C16 have been reported to be between 230-270 °C. These values are clearly higher than the processing temperatures used in this research.
6. The explanation on the "burnt" appearance of the CAP-C8 tensile test specimens should be reviewed (page 5, figure 2 and text below), in accordance with Point 5 raised just before, and taking into account the working temperatures for other composites using C8 FACE. If the thermogravimetric analyzes had been carried out on the FACEs before injection, this could have been anticipated by reducing (slightly) the injection temperature for example, below the degradation temperature of the C8 FACE degradation (if it is indeed a degradation of the C8 FACE of course)
Answer: Degradation temperatures of pure cellulose C8 and C16 have been reported to be between 230-270 °C that are clearly higher than the processing temperatures used in this research. The reason for the abnormal dark color of the CAP-C8 sample is not clear. No such strong color formation was observed in other compounds that also contained the C8 additive. However, the original color of C8 was more brownish than the color of C16. The high C8 amount, 17%, in translucent sample CAP-C8 may also partly cause visually dark outlook.
Reviewer 2 Report
The Authors have presented an interesting alternative for commonly used materials in 3D printing (FDM method). I have only three suggestions concerning paper modifications before publishing.
- Is it necessary to have the Results and Discussion section first, before the Materials and Methods? In most cases it is easier to understand when the results discussion comes after the conducted experiment was fully characterized.
- The Introduction section could use a short paragraph on the 3D printing methods available and the possible applications of the materials discussed here.
- When discussing the HDT (line 263) the results interpretation could use some improvement as it is in my opinion to much focused on the numbers per se.
The presented article "Thermoplastic cellulose-based compound for additive manufacturing" is a very well written document in which I have found only one typo: (line 120) after “Figure 1 3D” it feels as there is something missing, a dot or a comma.
Author Response
Reviewer 2:
The Authors have presented an interesting alternative for commonly used materials in 3D printing (FDM method). I have only three suggestions concerning paper modifications before publishing.
- Is it necessary to have the Results and Discussion section first, before the Materials and Methods? In most cases it is easier to understand when the results discussion comes after the conducted experiment was fully characterized.
Answer: We have used the template of this journal. The order of sections has been defined in this template and therefore "Materials and methods" section is after "Results and discussion" section.
2. The Introduction section could use a short paragraph on the 3D printing methods available and the possible applications of the materials discussed here.
Answer: Thank you for the comment. We have now included a short section describing 3D printing methods and possible applications in the Introduction.
3. When discussing the HDT (line 263) the results interpretation could use some improvement as it is in my opinion to much focused on the numbers per se.
Answer: Thank you for the comment. More discussion about HDT is now added.
The presented article "Thermoplastic cellulose-based compound for additive manufacturing" is a very well written document in which I have found only one typo: (line 120) after “Figure 1 3D” it feels as there is something missing, a dot or a comma.
Answer: Good remark. A dot is now added.
Round 2
Reviewer 1 Report
I Would like to thank the authors for their answers and explainations.